# Preoperative exercise training for adults undergoing elective major vascular surgery: A systematic review

**Garry A. Tew**[1]*, **Kim Caisley**[1], **Gerard Danjoux**[2]

1 Department of Sport, Exercise and Rehabilitation, Northumbria University, Newcastle upon Tyne, United Kingdom, 2 Department of Academic Anaesthesia, James Cook University Hospital, Middlesbrough, United Kingdom

* garry.tew@northumbria.ac.uk

**Data Availability Statement:** All relevant data are within the paper and its Supporting Information files.

**Funding:** The author(s) received no specific funding for this work.

## Abstract

Patients undergoing major vascular surgery may have an increased risk of postoperative complications due to poor 'fitness for surgery'. Prehabilitation aims to optimise physical fitness and risk factors before surgery to improve outcomes. The role of exercise-based prehabilitation in vascular surgery is currently unclear. Therefore, the aim of this systematic review was to assess the benefits and harms of preoperative exercise training in adults undergoing elective vascular surgery. We searched MEDLINE, Embase, CINAHL, and CENTRAL databases, trial registries, and forward and backward citations for studies published between January 2008 and April 2021. We included randomised trials that compared patients receiving exercise training with those receiving usual care or no training before vascular surgery. Outcomes included mortality, complications, and health-related quality of life (HRQOL). Three trials with 197 participants were included. All studies involved people undergoing abdominal aortic aneurysm (AAA) repair. Low-certainty evidence could not differentiate between rates of all-cause mortality. Moderate-certainty evidence indicated that postoperative cardiac and renal complications were less likely to occur in people who participated in preoperative exercise training compared with those who did not. Low-certainty evidence also indicated better postoperative HRQOL outcomes in people who undertook prehabilitation. There were no serious exercise-related adverse events. The evidence on preoperative exercise training for AAA patients is promising, but currently insufficiently robust for this intervention to be recommended in clinical guidelines. High-quality trials are needed to establish its clinical and cost-effectiveness. Research is also needed to determine the feasibility and effects of prehabilitation before lower-limb revascularisation.

**Trial registration:** PROSPERO ID: CRD42021245933.

## Introduction

Major vascular surgeries, such as lower-limb revascularisation, abdominal aortic aneurysm (AAA) repair, and carotid endarterectomy (CEA), are increasingly common and carry a

**Competing interests:** The authors of this manuscript have the following competing interests: GT and GD were investigators on one of the included trials (Tew 2017). This does not alter our adherence to PLOS ONE policies on sharing data and materials.

relatively high risk of perioperative mortality and major complications. In the United Kingdom (UK), 18,090 lower-limb bypass procedures were carried out in 2017–2019 [1]. For elective procedures (n = 11,283 [62%]), 1.0% of patients died in hospital and 10.2% were readmitted within 30 days. The same report stated that 3,445 elective infra-renal AAA repairs and 4,141 CEAs were performed in 2019 [1]. For patients undergoing open AAA repair (n = 1,355 [39%]), 2.3% died in hospital and 4.7% were readmitted within 30 days. For CEA, the corresponding figures were 1.9% and 4.4%, respectively.

Vascular procedures are complicated by the common prevalence amongst patients of comorbidities such as hypertension, diabetes, chronic lung disease and ischaemic heart disease, and other perioperative risk factors such as advanced age, low physical fitness, and smoking. For example, in relation to the elective lower-limb bypass procedures mentioned above, 47.1% of patients were over the age of 70 years, 31% were current smokers, 70% had hypertension, and 33.7% had ischaemic heart disease [1]. Such factors can reduce a patient's ability to withstand the physiological stress of major surgery, which is a key determinant of outcome. With lower 'fitness for surgery', the risk of complications and readmissions increases, and more intensive postoperative care is typically required [2–6]. The optimisation of modifiable risk factors in the weeks between diagnosis and treatment is therefore important for improving postoperative outcomes, and is an area of growing interest [7,8].

"Prehabilitation" is a term used to describe structured preoperative interventions to increase physiological reserve and address modifiable risk factors [9,10]. It typically includes three stages: screening/assessment, individualised needs-based intervention(s), and post-treatment evaluation. Exercise training is a common feature of prehabilitation programmes and is used either alone or as part of a multimodal intervention involving one or more of the following: nutritional support, psychological support, smoking cessation, alcohol reduction, and management of comorbidities. Preoperative exercise programmes may include aerobic training, resistance training, respiratory muscle training, or varying combinations of these activities.

In cancer care, prehabilitation has been shown to produce meaningful improvements in perioperative risk factors within two weeks [11], thereby facilitating patient readiness for surgery without undue delay. Subsequent benefits include a reduced risk of postoperative complications [12], better functional capacity [13], and health and social care financial savings [14]. Such findings have supported the development of principles and guidelines on prehabilitation for people with cancer [15] and new cancer prehabilitation services [16]. The role of prehabilitation for major vascular surgery is, however, less clear. For example, an evidence review for the 2020 National Institute for Health and Care Excellence (NICE) guideline on the diagnosis and management of AAA concluded that the evidence on preoperative exercise training was not robust enough to support a recommendation [17].

Although evidence on prehabilitation has been reviewed in the context of specific, individual vascular procedures (e.g., AAA repair [18]), no comprehensive review is available that covers all the most common, higher-risk procedures (i.e., lower-limb revascularisation, AAA repair, and CEA). We therefore conducted a systematic review to evaluate the benefits and harms of prehabilitation including exercise compared with usual care or no prehabilitation on pre- and post-operative outcomes in people undergoing elective major vascular surgery.

## Methods

This systematic review was conducted in accordance with the methods described in the Cochrane Handbook for Systematic Reviews of Interventions v6.2 [19], reported according to the Preferred Reporting Items for Systematic Reviews and Meta-analysis (PRISMA) 2020

statement [20], and registered with the International Prospective Register of Systematic Reviews (PROSPERO 2021 CRD42021245933).

## Study selection

Randomised controlled trials (RCTs) and quasi-RCTs (such as those that allocate participants to groups based on location of residence or date of assessment) of exercise training for adults (age ≥18 years) preparing for elective major vascular surgery that were published between January 1, 2008 and April 26, 2021 and met the inclusion criteria (Table 1) were identified by using our predefined search criteria (S1 Text) within the following databases: MEDLINE (Ovid), Embase (Ovid), CINAHL Complete (EBSCO), CENTRAL. The database searches were rerun on December 7, 2021. The surgical populations were limited to peripheral artery disease (PAD), AAA, and carotid artery disease to maintain a focus on vascular procedures (i.e., lower-limb revascularisation, AAA repair, and CEA, respectively) for which prehabilitation might be of greater importance. The review was restricted to studies published from 2008 onward because that was the year of the earliest study [21] included in previous reviews of exercise training before vascular surgery [22,23]. Relevant studies were also sought through screening of trial registries (ClinicalTrials.gov and ICTRP) and forward and backward citations of included studies.

Search results were imported into EndNote X9. Two authors (GT, KC) independently screened the identified records based on their title and abstract. When there was insufficient

**Table 1. Inclusion criteria.**

| Category | Description |
|---|---|
| Design | RCTs or quasi-RCTs |
| Population | Adults (age ≥18 years) scheduled to undergo elective procedures for PAD, AAA, or carotid artery disease |
| Intervention | The offer of structured, preoperative exercise training (≥7 days' duration) either alone or as part of a multimodal intervention. Exercise training could be aerobic training, resistance training, respiratory muscle training, or any combination of these activities. |
| Comparator | Usual care or no exercise training |
| Outcome measures | At least one of the following needed to be measured: Postoperative mortality (at 30 days and maximum follow-up)<br>• Postoperative complication rate (e.g., assessed using the Clavien Dindo scale or Comprehensive Complication Index)<br><br>• Health-related quality of life (e.g., assessed using the SF-36, EQ-5D or a disease-specific tool such as the VascuQoL)<br><br>• Amputation-free survival in revascularisation procedures<br><br>• Hospital readmission<br><br>• Length of hospital and critical care stay<br><br>• Functional capacity (e.g., 6-minute walk distance, peak oxygen consumption)<br><br>• Psychological health (e.g., anxiety, depression, or stress; assessed using a validated questionnaire)<br><br>• Adverse events related to exercise<br><br>• Adherence to the exercise programme |

AAA, abdominal aortic aneurysm; PAD, peripheral artery disease; RCT, randomised controlled trial

information to determine eligibility, full texts were retrieved and screened. Disagreements about study eligibility were resolved through discussion.

## Data extraction and risk of bias assessment

Two authors (GT, KC) independently extracted data from the included studies using a standardised form in Microsoft Excel. Extracted data were compared, with any discrepancies being resolved through discussion. The data extracted from each study included author names, publication date, country, study design, participant eligibility criteria and baseline characteristics, outcome measures (Table 1) and times assessed, length of follow up, and funding. Primary outcomes were postoperative mortality, postoperative complications, and health-related quality of life (HRQOL).

Prehabilitation intervention details were also extracted. These included intervention timeframes, components of multimodal interventions, and details of the exercise component according to the Consensus on Exercise Reporting Template (CERT) [24]. The CERT, which was designed to support the complete reporting of exercise programmes, comprises 16 items listed under 7 sections/domains: what (materials); who (provider); how (delivery); where (location); when, how much (dosage); tailoring (what/how); how well (compliance/planned and actual).

We assessed risk of bias in the included studies using the revised Cochrane Risk of Bias tool for randomised trials (RoB 2.0) [25]. RoB 2.0 addresses five specific domains: bias arising from the randomisation process; bias due to deviations from intended interventions; bias due to missing outcome data; bias in measurement of the outcome; and bias in selection of the reported result. Two authors (GT, KC) independently applied the tool to each included study for each of the three primary outcomes and recorded supporting information and justifications for judgements of risk of bias for each domain (low; high; some concerns). Any discrepancies in judgements of risk of bias or justifications for judgements were resolved by discussion. Following guidance [25], we derived an overall summary risk of bias judgement (low; some concerns; high) for each outcome, whereby the overall risk of bias for each study was determined by the highest risk of bias level recorded across the domains.

## Data synthesis and analysis

We synthesised the data in both narrative and tabular formats. Although we planned meta-analyses, the included studies provided varying outcomes and data that could not be combined in a meta-analysis. We therefore did not perform a meta-analysis. We present summary outcome data and effect estimates as reported in the original trial reports.

Two authors (GT, KC) independently assessed the certainty of the evidence using the GRADE approach [26]. When applicable, we followed published guidance for rating the certainty in evidence in the absence of a single estimate of effect [27]. We assessed the certainty of evidence for a particular outcome as high, moderate, low, or very low. We used GRADEpro software to prepare a 'Summary of findings' table [28]. We justified all decisions to down- or up-grade the certainty of evidence using footnotes.

## Results

The database searches yielded 23,113 records. After removing duplicates, we screened 18,506 records, from which we reviewed seven full-text documents, and finally included three studies [21,29,30]. Later, we reviewed the forward and backward citations of these three included studies. This resulted in a further seven full-text documents being reviewed; however, none of

PRISMA 2020 flow diagram for new systematic reviews which included searches of databases, registers and other sources

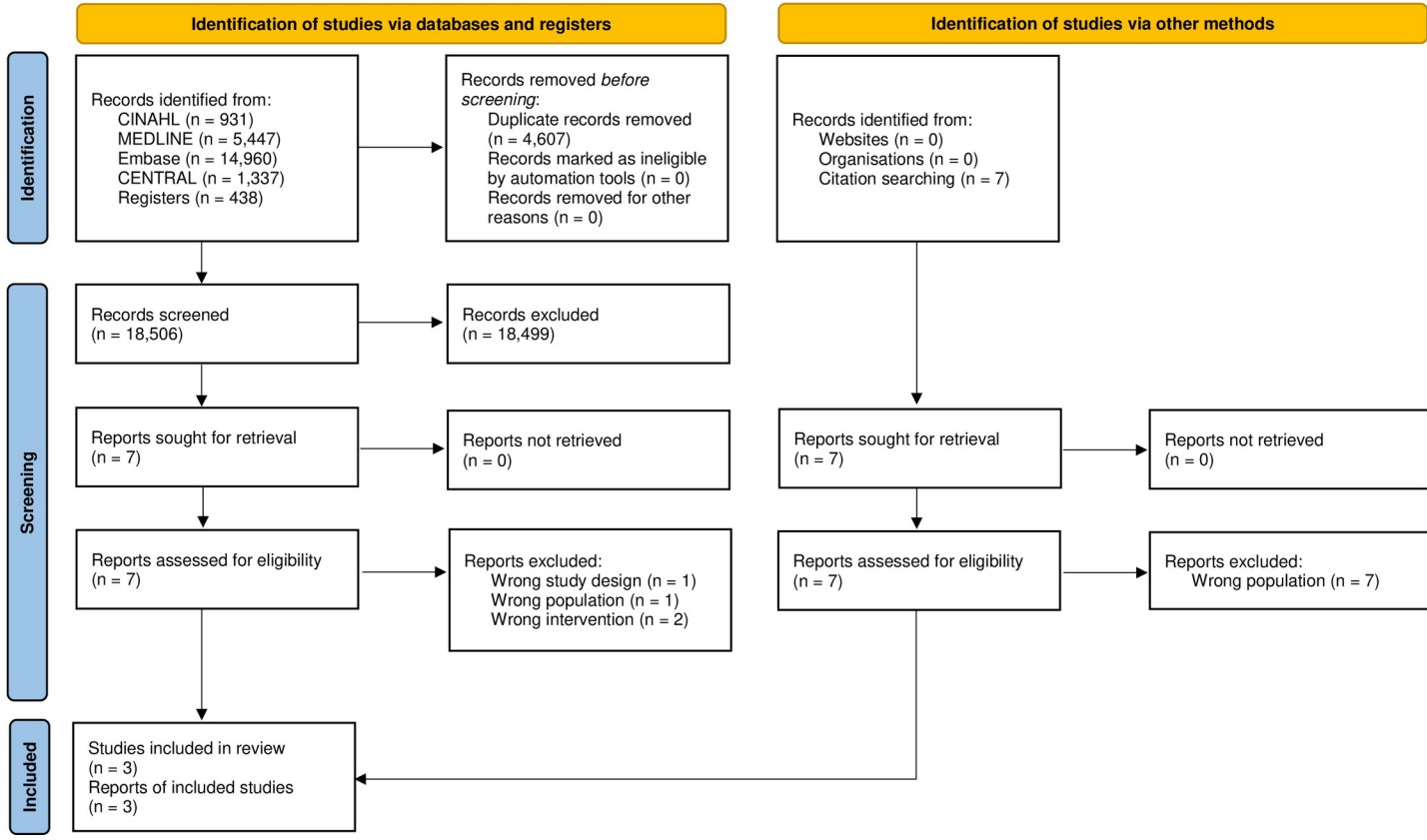

**Fig 1. PRISMA flow diagram.**

these fulfilled the inclusion criteria. The records that were excluded in the full-text reviewing stage are listed in S1 Table. Fig 1 shows the PRISMA flow diagram.

## Study characteristics

Table 2 summarises the characteristics of the three included studies. All studies were parallel-group trials where participants were individually randomised either to an experimental group that received preoperative exercise training or a control group that received usual care. One study was conducted in the Netherlands [21], and the other two in the UK [29,30]. All three studies involved people undergoing AAA repair; a mixture of open and endovascular repairs in two studies [29,30], and unspecified in the other [21]. A total of 197 participants were included in these studies, with study sample sizes ranging from 20 to 124. Group mean ages ranged from 59 to 75 years across the studies. In two studies [29,30] participants were mostly male, and in the third study [21] they were mostly female.

Detailed descriptions of the exercise interventions are provided in S2 Table. In one study [21], the exercise comprised of inspiratory muscle training, with participants performing one 15-minute exercise session, 6 days a week, for at least 2 weeks prior to surgery. In another study [29], participants attended hospital-based exercise classes three times per week for 6 weeks preoperatively. The classes used a circuit format with a mixture of aerobic and resistance exercises used at the exercise stations. In the third study [30], participants attended hospital-based exercise sessions three times per week for 4 weeks preoperatively, with weekly

**Table 2. Study characteristics.**

| Study ID | Study design | Population | Experimental groups | | Outcomes | | | |
|---|---|---|---|---|---|---|---|---|
| | | | Intervention | Control | Mortality | Complications | HRQOL | Other |
| Dronkers 2008 [21] | Parallel-group, individually randomised RCT Follow-up: 7 days after surgery | Country: Netherlands Procedure: AAA repair (not specified) Sample size: 20 | Sample size: 10 Mean age (y): 70 Male: 20% | Sample size: 10 Mean age (y): 59 Male: 30% | ✓ | ✓ Pulmonary | ✗ | Adverse events Adherence |
| Barakat 2016 [29] | Parallel-group, individually randomised RCT Follow-up: 3 months after discharge | Country: UK Procedure: AAA repair (open repair or EVAR) Sample size: 124 | Sample size: 62 Mean age (y): 74 Male: 90.3% EVAR: 37.1% | Sample size: 62 Mean age (y): 73 Male: 88.7% EVAR: 37.1% | ✓ | ✓ Cardiac, pulmonary, renal, SIRS, bleeding requiring reoperation or transfusion | ✗ | Length of stay Preoperative fitness (VT, peak $\dot{V}O_2$) Adverse events Adherence |
| Tew 2017 [30] | Parallel-group, individually randomised RCT Follow-up: 12 weeks after discharge | Country: UK Procedure: AAA repair (open repair or EVAR) Sample size: 53 | Sample size: 27 Mean age (y): 75 Male: 92.6% EVAR: 59.3% | Sample size: 26 Mean age (y): 75 Male: 96.2% EVAR: 57.7% | ✓ | ✓ Organ-specific: POMS | ✓ EQ-5D, SF36-PF, SF36-MH | Readmissions Length of stay Preoperative fitness (VT, peak $\dot{V}O_2$) Adverse events Adherence |

AAA, abdominal aortic aneurysm; EQ-5D, EuroQol 5-dimensions questionnaire; EVAR, endovascular aneurysm repair; HRQOL, health-related quality of life; peak $\dot{V}O_2$, peak oxygen consumption; POMS, Post-Operative Morbidity Survey; RCT, randomised controlled trial; SF36-MH, 36-Item Short Form Health Survey–Mental Health subscale; SF36-PF, 36-Item Short Form Health Survey–Physical Functioning subscale; SIRS, systemic inflammatory response syndrome; VT, ventilatory threshold.

'maintenance' sessions included if the operation was delayed. Each session involved high-intensity interval training on a cycle ergometer. None of the studies reported the concomitant use of any other prehabilitation interventions (e.g., smoking cessation, nutritional support).

Outcome measures varied across the included studies. All three studies reported data on postoperative mortality and complications. Postoperative complications were assessed by a blinded investigator using a variety of methods. One study assessed pulmonary complications as the incidence of atelectasis in the 7 days after surgery [21]. Another study [30] assessed organ-specific morbidity using the Post-Operative Morbidity Survey (POMS), reporting the total POMS count up to the point of hospital discharge. The third study [29] assessed the incidence of cardiac, pulmonary, and renal complications (according to specified definitions) as a composite primary outcome, as well as the incidence of systemic inflammatory response syndrome and postoperative bleeding requiring reoperation or transfusion. Only one study [30] assessed HRQOL, with participants self-completing the SF-36v2™ Health Survey to determine scores for the physical functioning and mental health subscales, and the EQ-5D-5L to determine scores for the utility index and visual analogue scale (VAS). Other postoperative outcomes included readmissions [30] and length of hospital stay [29,30]. Two studies [29,30] measured the change in cardiorespiratory fitness before surgery by determining ventilatory threshold and peak oxygen consumption from cardiopulmonary exercise testing. All studies reported data on exercise-related adverse events. All studies also reported data on intervention adherence: in one study [21] participants documented their training sessions in a diary, in another study [29] it appeared that registers were used to document class attendance, and in

the third study [30] an investigator completed a case report form to document session attendance and acute responses to exercise (e.g., heart rate, blood pressure, perceived exertion).

## Outcome data

Table 3 summarises the data on primary outcome measures, adverse events, and adherence from each study. Two studies [21,30] reported no postoperative deaths during follow-up and one study [29] reported two deaths in each of the two study groups. Combining the studies

**Table 3. Outcome data.**

| Study ID | Mortality[a] | Complications | HRQOL | Adverse events | Adherence |
|---|---|---|---|---|---|
| Dronkers 2008 [21] | 0 | Incidence of pulmonary complications: Intervention: 3/10 Control: 8/10 RR 0.38 (0.14, 1.02)[b] | N/A | 0 | "All participants reported their daily inspiratory muscle training workout in their diaries." |
| Barakat 2016 [29] | 2 in each group | Incidence of cardiac complications: Intervention: 5/62 (EVAR 1, OAR 4) Control: 14/62 (EVAR 3, OAR 11) RR 0.36 (0.14, 0.93)[b] Incidence of pulmonary complications: Intervention: 7/62 (EVAR 0, OAR 7) Control: 13/62 (EVAR 4, OAR 9) RR 0.54 (0.23, 1.26)[b] Incidence of renal complications: Intervention: 4/62 (EVAR 1, OAR 3) Control: 13/62 (EVAR 1, OAR 12) RR 0.31 (0.11, 0.89)[b] | N/A | 0 | 0 classes: 11/62 6–12 classes: 19/62 13–18 classes: 32/62 |
| Tew 2017 [30] | 0 | Total POMS count up to hospital discharge: Intervention (mean): 2.3 Control (mean): 2.1 (difference 0.2, 95% CI -0.3 to 0.7) | EQ-5D utility index scores at 12 weeks: Intervention (mean, n = 21): 0·837 Control (mean, n = 22): 0·760 (difference 0·077, 0·005 to 0·148) EQ-VAS score at 12 weeks: Intervention (mean, n = 21): 79·6 Control (mean, n = 22): 74·4 (difference 5·2, –1·7 to 12·0) SF-36 physical functioning at 12 weeks: Intervention (mean, n = 22): 49·4 Control (mean, n = 21): 46·5 (difference 2·9, 0·4 to 5·4) SF-36® mental health at 12 weeks: Intervention (mean, n = 21): 55·6 Control (mean, n = 21): 55·0 (difference 0·6, –2·4 to 3·6) | 2 (dizziness, angina) | 240/324 (74%) main-phase and 36/40 (90%) maintenance sessions completed (overall attendance = 75·8%) 17/27 (63%) attended ≥9/12 main-phase sessions and all maintenance sessions 30% of exercise bouts were reported to be in the target range of hard to very hard |

EVAR, endovascular aneurysm repair; HRQOL, health-related quality of life; OAR, open aneurysm repair; POMS, Post-Operative Morbidity Survey; RR, risk ratio.

[a]At 7 days after surgery for Dronkers 2008 and 30 days after surgery for Barakat 2016 and Tew 2017.

[b]Risk ratios taken from [17]. Effect sizes below 1 favour the intervention group.

gave a death rate of 1.9% in each group. A meta-analysis was not performed due to the low number of events and heterogeneity of interventions. Regarding complications, the largest trial [29] provided evidence that cardiac and renal complications were less likely to occur in people who participated in preoperative hospital-based exercise classes compared with those who had not. An exploratory (i.e., underpowered) sub-group analysis indicated that the effect was procedure-specific, with a greater risk reduction being observed in patients undergoing open surgical repair rather than endovascular repair. The same trial, and an earlier pilot study [21], could not differentiate between rates of pulmonary complications. A meta-analysis was not performed due to heterogeneity in interventions and outcome assessment. The one trial [30] that measured HRQOL reported superior scores in the intervention group at 12 weeks after discharge for EQ-5D utility index and SF-36 physical functioning; however, differences in EQ-VAS and SF-36 mental health scores were not statistically significant. There were no serious exercise-related adverse events. One study [30] reported two non-serious adverse events: dizziness and angina. It appeared that adherence to exercise was generally good; how-ever, in the study of Tew et al. [30] the intensity of exercise was generally lower than intended, with only 30% of exercise bouts being reported to be in the target range of hard to very hard.

Two studies [29,30] reported data on length of hospital stay. For intervention versus con-trol, median length of hospital stay was 7 versus 8 days (P = 0.025), respectively in the study of Barakat et al. [29] and 7 versus 6 days (P not reported) in the study of Tew et al. [30]. The same two studies also used cardiopulmonary exercise testing to assess ventilatory threshold and peak oxygen consumption (measures of cardiorespiratory fitness) at baseline and preopera-tively (after prehabilitation). In the study of Barakat et al. [29], there were statistically signifi-cant (P<0.05) improvements in ventilatory threshold and peak oxygen consumption in the intervention group (mean changes of +1.9 and +1.6 ml/kg/min, respectively) but not in the control group (mean changes of -0.2 and -1.2 ml/kg/min, respectively). In the study of Tew et al. [30], preoperative scores for these outcomes did not differ significantly between the two groups (mean differences of 0·3 [95% CI -0·4 to 1·1] and 0·5 [95% CI -0·6 to 1·7] ml/kg/min, respectively). One study [29] reported data on the incidence of systemic inflammatory response syndrome (SIRS) and bleeding requiring reoperation or transfusion. During hospital stay, SIRS occurred in 50/62 (80.6%) intervention participants and 51/62 (82.3%) control par-ticipants (P = 0.817). Rates of postoperative bleeding were 4/62 (6.5%) and 7/62 (11.3%), respectively (P = 0.343). Finally, one study [30] reported data on hospital readmissions: 0/27 (0%) versus 3/26 (11.5%) for intervention and control groups, respectively.

### Risk of bias and certainty of evidence

A summary of the risk of bias assessments is provided in Fig 2. For the outcomes of postopera-tive mortality and complications, the overall risk of bias was judged to be low in all three studies. For HRQOL, the study of Tew et al. [30] received an overall rating of 'some concerns' because participants completed the questionnaires with knowledge of the intervention received.

The certainty of evidence assessments for the primary outcomes are summarised in Table 4. The evidence was rated as moderate for cardiac and renal complications, and low for mortality, pulmonary complications, and HRQOL. All outcomes were downgraded one or two levels for imprecision. HRQOL was also downgraded one level for its 'some concerns' risk of bias rating.

## Discussion
### Main findings

This systematic review evaluating preoperative exercise training in the context of major vascu-lar surgery identified three comparative studies in patients undergoing elective AAA repair

**Fig 2. Risk of bias judgements.**

and no studies in patients receiving any other vascular procedure. One study provided moderate-certainty evidence that postoperative cardiac and renal complications were less likely to occur in patients with an AAA who participated in preoperative exercise training compared with those who did not. Another study provided low-certainty evidence of better postoperative HRQOL outcomes in people who had received exercise. Rates of all-cause mortality and pulmonary complications could not be differentiated between groups (low-certainty evidence). There were no serious exercise-related adverse events. This review provides researchers, healthcare practitioners, and policy makers with an overview of the current evidence to inform future research directions and clinical practice.

## Exercise training before AAA repair

We identified three RCTs that provided data on a broad range of outcome measures (Tables 2–4). Overall, our impression is that the current evidence on efficacy and safety is promising,

**Table 4. Summary of findings.**

| Outcomes | Impact | Number of participants (studies) | Certainty of the evidence (GRADE) |
|---|---|---|---|
| Postoperative mortality | The studies could not differentiate between rates of all-cause mortality. | 197 (3 RCTs) | ⊕⊕◯◯ LOW [a] |
| Cardiac complications (including myocardial infarction, prolonged inotropic support, new onset arrhythmia, and unstable angina) | One study reported a lower rate of complications in the intervention group. | 124 (1 RCT) | ⊕⊕⊕◯ MODERATE [b] |
| Pulmonary complications (including atelectasis, pneumonia, pneumonia requiring reintubation, exacerbation of COPD, and reintubation) | Two studies could not differentiate between rates of complications. | 144 (2 RCTs) | ⊕⊕◯◯ LOW [a] |
| Renal complications (including acute renal failure and renal insufficiency) | One study reported a lower rate of complications in the intervention group. | 124 (1 RCT) | ⊕⊕⊕◯ MODERATE [b] |
| Postoperative HRQOL (including SF-36 physical functioning and mental health subscales and EQ-5D utility index) | One study reported better postoperative HRQOL outcomes in the intervention group. | 53 (1 RCT) | ⊕⊕◯◯ LOW [b,c] |

Explanations

a. Confidence interval crosses two lines of a defined minimum clinically important difference (for complications: RR MIDs of 0.8 and 1.25), downgrade 2 levels.

b. Confidence interval crosses one line of a defined minimum clinically important difference (for complications: RR MIDs of 0.8 and 1.25), downgrade 1 level.

c. Participants completed the questionnaires with knowledge of the intervention received, downgrade 1 level.

but insufficiently robust for preoperative exercise training to be recommended in clinical guidelines. For example, the small sample sizes and relatively short follow-up periods reduces confidence in the reported outcomes. Our findings are consistent with three other recent reviews of preoperative exercise for people with AAA [17,18,31], which identified the same primary studies. These reviews all called for further research. Our search of trial registries identified three relevant ongoing studies that will include a combined total of 195 participants [32–34]. However, these studies all have a main focus on feasibility, so an adequately powered RCT would still be needed to establish clinical and cost-effectiveness. Suggested features of a full-scale trial are as follows:

- Population: adults undergoing elective repair of an unruptured AAA

- Intervention: a supervised exercise programme that prioritises aerobic exercise training but may also include resistance training and respiratory muscle training. In the context of the UK healthcare system, the programme duration should be 4–6 weeks to fit with the 8-week referral to repair target.

- Comparator: usual care (including preoperative assessment but no structured exercise)

- Outcomes: postoperative mortality and complications, HRQOL, adverse events, resource use and costs

- Study design: pragmatic multi-centre RCT

Whether or not to include people undergoing EVAR will require careful consideration; their inclusion would facilitate recruitment, but possibly at the expense of reducing the effect size (preoperative fitness is less important before EVAR because this procedure is less metabolically demanding than open surgery). Indeed, the study of Barakat et al. [29] provided preliminary data indicating a greater effect of prehabilitation on postoperative complications in people undergoing open surgery compared with those receiving EVAR. Further research is necessary to better understand which patient sub-groups stand to benefit most from prehabilitation. Other potential areas of future research include the comparative effectiveness of different exercise programmes, the feasibility and effects of delivering exercise as part of a multimodal prehabilitation programme, the role of new technologies and remote modes of delivery, and the values and preferences of patients regarding prehabilitation options.

There is limited direct evidence to support the recommendation of a specific prehabilitation programme for people undergoing elective AAA repair (or any other vascular procedure). Guidelines from the National Institute for Health and Care Excellence (NICE) on improving surgical outcomes for people with AAA [35] recommend that exercise be promoted to improve cardiovascular and general health, however specific programme details (e.g., mode, frequency, intensity, duration) are not provided, and individuals are signposted to other NICE guidelines on physical activity that are not specific to people with AAA [36]. Other guidelines provide more specific recommendations on preoperative exercise training [37,38], but these are largely based on data from other surgical populations and expert opinion. Notable recommendations are as follows:

- A stepped care model of prehabilitation should be considered, where higher risk patients receive targeted and more intensive individualised interventions and low risk patients receive more generalised universal interventions (e.g., preoperative education)

- A combination of aerobic training, resistance training, and respiratory muscle training should be delivered, however aerobic training should be prioritised because aerobic exercise capacity specifically is associated with surgical outcome [4–6]

- Moderate to high intensity interval exercise programmes are recommended

- Exercise training should commence as early in the surgical pathway as possible

- Supervised exercise programmes are preferred using either face-to-face or remote/virtual supervision

Regarding safety considerations, again there is little direct evidence to inform practice. However, given the nature of the population, we would advocate a cautious approach. Pragmatic control measures include thorough preparticipation screening, exclusion of patients with contraindications to exercise (e.g., severe aortic stenosis, uncontrolled arrhythmia), exercise sessions supervised and delivered by individuals with relevant expertise, prompt evaluation of untoward medical signs and symptoms, first aid equipment immediately available, and reduction in exercise intensity if a patient has systolic blood pressure rise to >180 mmHg or heart rate >95% of their maximum [39]. The latter point relates to a concern that excessive rises in double product (systolic blood pressure × heart rate) during excessive training may evoke aneurysm expansion or rupture. However, the available evidence suggests that these concerns are unfounded [40].

## Exercise training before lower-limb revascularisation for PAD

We found no eligible studies of preoperative exercise training in patients undergoing lower-limb revascularisation for PAD; a finding that is consistent with a recent Cochrane review on this topic [22]. However, other evidence (mostly indirect) highlights the potential benefit of prehabilitation in this specific population: (i) observational data showing that impaired functional capacity is associated with an increased risk of postoperative complications after infrainguinal bypass surgery [41]; (ii) meta-analyses of RCTs showing that structured exercise training can produce meaningful improvements in functional capacity in people who are being managed conservatively for intermittent claudication due to PAD [42,43], and (iii) meta-analyses of RCTs showing that preoperative exercise training can reduce postoperative complications in patients undergoing major abdominal surgery [44,45]. However, we would caution against extrapolating from this evidence due to some unique features of this population. For example, ischaemic leg pain owing to the condition might make 'conventional' exercise programmes based on walking, running, or cycling difficult to perform. Accordingly, further research is needed to generate evidence that is specific to this population.

Our search of trial registries failed to identify any relevant ongoing studies, but we did identify one study that had been terminated early due to "Difficulty with enrolment and change in available study resources" [46]. This pilot RCT sought to ascertain the feasibility and acceptability of recruiting people with lower-extremity PAD to a prehabilitation programme in which participants were encouraged to increase their physical activity, practice stress reduction, and engage in other healthy behaviours "in the days leading up to surgery". The primary outcomes were the feasibility and acceptability of participation, with specific targets including the enrolment of 25–40 participants, a drop-out rate of <15% at the 8-month follow-up, and >85% usage of the pedometers by participants in the intervention group. The study was stopped after having only enrolled seven patients in 16 months. The exclusion of patients with critical limb ischaemia may have hampered recruitment by vastly reducing the number of potentially eligible patients, however the outcome still indicates that prehabilitation trials in the PAD population might be difficult to perform. Accordingly, we recommend that further work is done to establish the feasibility of the intervention and key trial processes (e.g., recruitment) prior to embarking on a full-scale trial. Suggested features of a feasibility study are as follows:

- Population: adults undergoing elective lower-limb revascularisation, including people with intermittent claudication with lifestyle limiting symptoms for whom conservative treatment did not work and people with critical limb ischaemia

- Intervention: a supervised exercise programme that prioritises aerobic exercise training but may also include resistance training and respiratory muscle training

- Comparator: usual care (including preoperative assessment but no structured exercise)

- Outcomes: feasibility–rates of recruitment, retention, adherence, outcome completion; other–postoperative mortality and complications, HRQOL, adverse events, resource use and costs

- Study design: pragmatic multi-centre pilot RCT

Whether or not to include endovascular procedures requires similar considerations to those discussed previously for the AAA population. The exercise intervention also needs further development since lower-body exercise may not be acceptable to many patients. Programmes involving arm ergometry may be worth exploring because this mode of training has been shown to improve aerobic fitness and walking distances in people with intermittent claudication [47].

## Exercise training before revascularisation for carotid artery disease

We also did not find any completed or ongoing trials of preoperative exercise training in carotid artery disease. This is perhaps unsurprising for two main reasons. Firstly, conducting a trial on this population would be challenging because the vast majority of procedures (in the UK at least) are performed in patients who have experienced transient symptoms or a stroke, for whom the target is to operate with two weeks of initial symptoms [48]. Thus, the timeframe to recruit and train patients would be very short. Secondly, it is unclear how preoperative exercise training would reduce postoperative complications in this population. The complication rate is low and procedures do not generate a large inflammatory response. We are also not aware of any data demonstrating an association between poor fitness and an increased risk of postoperative complications after CEA. Therefore, we do not think that revascularisation for carotid artery disease is a good model to explore the potential benefits of preoperative exercise in future research. This is not to say that other aspects of preoperative optimisation are not important (e.g., smoking cessation), and exercise may still have an important role as part of optimal medical management in patients with asymptomatic carotid stenosis [49,50] and in rehabilitation after stroke [51].

## Strengths and limitations of this review

This review benefits from robust methods in keeping with established guidelines [19], including a registered protocol. Searches were comprehensive and two review authors conducted the record screening, data extraction, and risk of bias assessments independently. The exercise interventions were also reported according to clinical consensus guidelines [24]. However, this review has some limitations. First, we did not identify any eligible studies involving PAD or carotid artery disease populations. Second, there were only three eligible studies in the context of AAA repair, two of which were pilot or feasibility trials [21,30], and the other of which was powered on a composite endpoint [29]. All had relatively short follow-up periods. Third, we were unable to conduct meaningful meta-analyses due to limited data and heterogeneity of interventions and outcome measures.

## Conclusion

The aim of our systematic review was to assess the benefits and harms of preoperative exercise training in adults undergoing elective major vascular procedures. The evidence on preoperative exercise training for people undergoing AAA repair is promising, but currently insufficiently robust for this intervention to be recommended in clinical guidelines. More high-quality trials are needed to guide best practice and policy. Research is also needed to determine the feasibility and effects of exercise training in people awaiting lower-limb revascularisation.

## Supporting information

**S1 Table. Records that were excluded at the full-text reviewing stage.**
(DOCX)

**S2 Table. Descriptions of the exercise components used in prehabilitation interventions according to Consensus Exercise Reporting Template (CERT).**
(DOCX)

**S1 File. Database search strategies.**
(DOCX)

## Author Contributions

**Conceptualization:** Garry A. Tew, Kim Caisley.

**Data curation:** Garry A. Tew, Kim Caisley.

**Formal analysis:** Garry A. Tew, Kim Caisley.

**Methodology:** Garry A. Tew, Kim Caisley.

**Supervision:** Garry A. Tew.

**Writing – original draft:** Garry A. Tew.

**Writing – review & editing:** Kim Caisley, Gerard Danjoux.

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
