## [Decision Letter · Decision Letter 0]

6 Dec 2021

PONE-D-21-32818Preoperative exercise training for adults undergoing elective major vascular surgery: A systematic reviewPLOS ONE

Dear Dr. Tew,

Thank you for submitting your manuscript to PLOS ONE. After careful consideration, we feel that it has merit but does not fully meet PLOS ONE’s publication criteria as it currently stands. Therefore, we invite you to submit a revised version of the manuscript that addresses the points raised during the review process.

We look forward to receiving your revised manuscript.

Kind regards,

Athanasios Saratzis

Academic Editor

PLOS ONE

Journal Requirements:

[I have read the journal's policy and the authors of this manuscript have the following competing interests: GT and GD were investigators on one of the included trials (Tew 2017).]

Additional Editor Comments:

Thank you for submitting this interesting article. Some comments have been made by the expert Reviewers. Please address the comments raised if you are considering a re-submission.

Reviewers' comments:

Reviewer's Responses to Questions

**Comments to the Author**

1. Is the manuscript technically sound, and do the data support the conclusions?

Reviewer #1: Yes

Reviewer #2: Yes

2. Has the statistical analysis been performed appropriately and rigorously? 

Reviewer #1: Yes

Reviewer #2: Yes

3. Have the authors made all data underlying the findings in their manuscript fully available?

Reviewer #1: Yes

Reviewer #2: Yes

4. Is the manuscript presented in an intelligible fashion and written in standard English?

Reviewer #1: Yes

Reviewer #2: Yes

5. Review Comments to the Author

Reviewer #1: For transparency to the authors i have already reviewed this document and provided feedback which i believe has mainly been addressed. However i did not see the previous response to reviewer document. Some further considerations for review:

1. With regard to the RoB tool, i would consider an external reviewer as some of the papers are the authors own.

2. Consider a secondary search as the current one is nearing 8 months post completion.

3. This review only included 3 studies but a recently published Cochrane - identifies 4 (https://pubmed.ncbi.nlm.nih.gov/34236703/) please consider whether the missing study should be included in this review?

4. Regarding the complications in the Baraket paper these were found to be surgery specific OPEN vs EVAR, it would be good to demonstrate this in the results section.

5. Page 15 lines 222-223 & 227-228 are repeats.

6. Please discuss the discrepancy between the RoB reported here as low but reported as very high in the Cochrane review.

7. Due to the lack of studies found in PAD and carotid disease, this review does have an overlap with the published Cochrane review. Is there consideration to expand your review to non-RCTs to include more data, although acknowledging that this is not as "high level evidence".

Reviewer #2: This is well-conducted systematic review of the published evidence regarding pre-habilitation exercise programmes prior to major vascular surgery (AAA, Lower limb bypasses and Carotid endarterectomy). Unfortunately there were only 3 RCTs which fulfilled the inclusion criteria, however this reflects the current lack of evidence in this area. The manuscript is well-written, however could benefit from shortening of the discussion section, where there seems to be a repetition of some of the findings in the results section

6. PLOS authors have the option to publish the peer review history of their article (what does this mean?). If published, this will include your full peer review and any attached files.

Reviewer #1: No

Reviewer #2: No

---

## [Author Response · Author response to Decision Letter 0]

7 Dec 2021

Response to Reviewers

Preoperative exercise training for adults undergoing elective major vascular surgery: A systematic review

Journal Requirements:

We have reformatted the headings and updated the figure citations and file names for figures. Supporting information has been uploaded separately as individual files. Postal codes have been removed from affiliations. 

[I have read the journal's policy and the authors of this manuscript have the following competing interests: GT and GD were investigators on one of the included trials (Tew 2017).]

We have re-named this section as “Competing interests” and inserted the following variant of what has been suggested: “The authors of this manuscript have the following competing interests: GT and GD were investigators on one of the included trials (Tew 2017).”

We have included that statement in the Competing Interests section. 

The updated statement has been included in our cover letter.

We have reviewed the reference list and believe it to be complete and correct.

Additional Editor Comments:

Thank you for submitting this interesting article. Some comments have been made by the expert Reviewers. Please address the comments raised if you are considering a re-submission.

We are glad that you think our article is interesting. We have addressed all review comments. 

Reviewers' comments:

Reviewer's Responses to Questions

Comments to the Author

1. Is the manuscript technically sound, and do the data support the conclusions?

Reviewer #1: Yes

Reviewer #2: Yes

2. Has the statistical analysis been performed appropriately and rigorously?

Reviewer #1: Yes

Reviewer #2: Yes

3. Have the authors made all data underlying the findings in their manuscript fully available?

Reviewer #1: Yes

Reviewer #2: Yes

4. Is the manuscript presented in an intelligible fashion and written in standard English?

Reviewer #1: Yes

Reviewer #2: Yes

5. Review Comments to the Author

Reviewer #1: For transparency to the authors i have already reviewed this document and provided feedback which i believe has mainly been addressed. However i did not see the previous response to reviewer document. Some further considerations for review:

We appreciate the time you have taken to review our manuscript and thank you for the thoughtful and helpful comments.

1. With regard to the RoB tool, i would consider an external reviewer as some of the papers are the authors own.

The risk of bias assessments were conducted independently by two reviewers. One of the reviewers (KC) was not involved in the any of the included studies. There were no discrepancies in two reviewers’ judgements, so we are confident that our assessments are appropriate and unbiased.

2. Consider a secondary search as the current one is nearing 8 months post completion.

We performed a rerun of the database searches on December 7, 2021. No further relevant RCTs were identified. 

3. This review only included 3 studies but a recently published Cochrane - identifies 4 (https://pubmed.ncbi.nlm.nih.gov/34236703/) please consider whether the missing study should be included in this review?

We think that the reviewer is referring to the following pilot study of Richardson and colleagues: https://clinicaltrials.gov/ct2/show/NCT02845167. This study was excluded during our screening process because the intervention involved exercise sessions on three consecutive days. For inclusion in our review, the training duration needed to be at least one week. 

4. Regarding the complications in the Baraket paper these were found to be surgery specific OPEN vs EVAR, it would be good to demonstrate this in the results section.

We have added the procedure-specific complication rates to Table 3, as well as the following sentence: “An exploratory (i.e., underpowered) sub-group analysis indicated that the effect was procedure-specific, with a greater risk reduction being observed in patients undergoing open surgical repair rather than endovascular repair.”

5. Page 15 lines 222-223 & 227-228 are repeats.

These sentences are worded similarly, but both are relevant and worthy of inclusion as they relate to different outcomes. The first sentence refers to how we handled the mortality data. The second sentence refers to how we handled the complications data. 

6. Please discuss the discrepancy between the RoB reported here as low but reported as very high in the Cochrane review.

The Cochrane review of Fenton et al. (2021) used version 1 of the Cochrane RoB tool, whereas we used version 2, so the RoB results are not directly comparable. We can however comment on some apparent discrepancies. First, we did not consider there to be high risk of bias due to selective reporting because data were presented for the outcome measures that we were interested in. Second, Fenton et al. gave the study of Tew et al. a rating of ‘high risk’ for incomplete outcome data, stating that attrition was >20% and ITT analysis wasn’t used. Both these statements are incorrect. Third, in ‘other potential sources of bias’, Fenton et al. gave a high risk of bias rating to the study of Tew et al. for it being underpowered. This goes against Cochrane guidance that criteria related to precision should not be assessed within this domain (https://handbook-5-1.cochrane.org/chapter_8/8_15_2_assessing_risk_of_bias_from_other_sources.htm). Finally, we would like to note that that RoB results in the NICE evidence review (reference #17) are in keeping with our results.

7. Due to the lack of studies found in PAD and carotid disease, this review does have an overlap with the published Cochrane review. Is there consideration to expand your review to non-RCTs to include more data, although acknowledging that this is not as "high level evidence".

Quasi-RCTs were eligible for inclusion, however no such studies were found. We decided not to include uncontrolled studies because they provide little insight into the quantitative effect of prehabilitation on post-operative outcomes. Limiting the inclusion of studies to RCTs also reflects the approach used in evidence syntheses to inform clinical guidelines (i.e., to focus on “high-level evidence). 

Reviewer #2: This is well-conducted systematic review of the published evidence regarding pre-habilitation exercise programmes prior to major vascular surgery (AAA, Lower limb bypasses and Carotid endarterectomy). Unfortunately there were only 3 RCTs which fulfilled the inclusion criteria, however this reflects the current lack of evidence in this area. The manuscript is well-written, however could benefit from shortening of the discussion section, where there seems to be a repetition of some of the findings in the results section.

Thank you for reviewing our manuscript. We have carefully reviewed the discussion section and would prefer not to shorten it. This is because we think that all the content is valuable and most of it is new (i.e., not repeated). For example, the discussion around implications for policy, practice and research and the strengths and limitations, which makes up most of the text, is all unique to this section. Also, at 1,828 words, the size of this section is in keeping with the rest of the manuscript.

---

## [Decision Letter · Decision Letter 1]

12 Jan 2022

Preoperative exercise training for adults undergoing elective major vascular surgery: A systematic review

PONE-D-21-32818R1

Dear Dr. Tew,

We’re pleased to inform you that your manuscript has been judged scientifically suitable for publication and will be formally accepted for publication once it meets all outstanding technical requirements.

Kind regards,

Athanasios Saratzis

Academic Editor

PLOS ONE

Additional Editor Comments (optional):

Reviewers' comments:

Reviewer's Responses to Questions

**Comments to the Author**

1. If the authors have adequately addressed your comments raised in a previous round of review and you feel that this manuscript is now acceptable for publication, you may indicate that here to bypass the “Comments to the Author” section, enter your conflict of interest statement in the “Confidential to Editor” section, and submit your "Accept" recommendation.

Reviewer #2: All comments have been addressed

2. Is the manuscript technically sound, and do the data support the conclusions?

Reviewer #2: Yes

3. Has the statistical analysis been performed appropriately and rigorously? 

Reviewer #2: N/A

4. Have the authors made all data underlying the findings in their manuscript fully available?

Reviewer #2: Yes

5. Is the manuscript presented in an intelligible fashion and written in standard English?

Reviewer #2: Yes

6. Review Comments to the Author

Reviewer #2: (No Response)

7. PLOS authors have the option to publish the peer review history of their article (what does this mean?). If published, this will include your full peer review and any attached files.

Reviewer #2: No

---

## [Editor Report · Acceptance letter]

17 Jan 2022

PONE-D-21-32818R1 

Preoperative exercise training for adults undergoing elective major vascular surgery: A systematic review 

Dear Dr. Tew:

I'm pleased to inform you that your manuscript has been deemed suitable for publication in PLOS ONE. Congratulations! Your manuscript is now with our production department. 

Kind regards, 

on behalf of

Dr. Athanasios Saratzis 

Academic Editor

PLOS ONE